# Development of the Hypertension Index Model in General Adult Using the Korea National Health and Nutritional Examination Survey and the Korean Genome and Epidemiology Study

**DOI:** 10.3390/jpm11100968

**Published:** 2021-09-28

**Authors:** Myung-Jae Seo, Sung-Gyun Ahn, Yong-Jae Lee, Jong-Koo Kim

**Affiliations:** 1Department of Family Medicine, Yonsei University Wonju College of Medicine, Wonju 26426, Korea; smjae@yonsei.ac.kr; 2Division of Cardiology, Department of Internal Medicine, Yonsei University Wonju College of Medicine, Wonju 26426, Korea; sgahn@yonsei.ac.kr; 3Department of Family Medicine, Yonsei University College of Medicine, Seoul 06273, Korea; ukyjhome@yuhs.ac

**Keywords:** hypertension, prediction model, cross-sectional, risk factors, sex-specific

## Abstract

Hypertension, a risk factor for cardiovascular disease and all-cause mortality, has been increasing. Along with emphasizing awareness and control of hypertension, predicting the incidence of hypertension is important. Several studies have previously reported prediction models of hypertension. However, among the previous models for predicting hypertension, few models reflect various risk factors for hypertension. We constructed a sex-specific prediction model using Korean datasets, which included socioeconomic status, medical history, lifestyle-related variables, anthropometric status, and laboratory indices. We utilized the data from the Korea National Health and Nutrition Examination Survey from 2011 to 2015 to derive a hypertension prediction model. Participants aged 40 years or older. We constructed a sex-specific hypertension classification model using logistic regression and features obtained by literature review and statistical analysis. We constructed a sex-specific hypertension classification model including approximately 20 variables. We estimated its performance using the Korea National Health and Nutrition Examination Survey dataset from 2016 to 2018 (AUC = 0.847 in men, AUC = 0.901 in women). The performance of our hypertension model was considered significant based on the cumulative incidence calculated from a longitudinal dataset, the Korean Genome and Epidemiology Study dataset. We developed this hypertension prediction model using features that could be collected in a clinical office without difficulty. Individualized results may alert a person at high risk to modify unhealthy lifestyles.

## 1. Introduction

Approximately 1.38 billion people worldwide were estimated to have hypertension in 2010, and this prevalence is estimated to increase by more than 1.6 billion by 2025 [1]. In Korea, the prevalence of hypertension was approximately 30.5% according to the 2007–2015 Korea National Health and Nutrition Examination Survey (KNHANES) [2]. Hypertension is the leading modifiable risk factor for cardiovascular disease (CVD), chronic kidney disease, and all-cause and CVD-related mortality [3,4,5].

The three major tasks for hypertension are high blood pressure (BP) awareness, treatment, and control [6]. Awareness and well-control of hypertension vary according to the region, sex, income, and education level [1]. Globally, the proportion of awareness is low, especially in middle- and low-income countries [1]. The ratio of patients with hypertension who received treatment was less than half, and the hypertension control ratio was only 13.8% globally in 2010 [1]. Kang et al. reported that 32.7% of patients with hypertension did not perceive their status [2]. Early diagnosis or prediction of hypertension is a crucial challenge because uncontrolled or untreated hypertension is known to increase mortality [5].

Two approaches have been attempted in the early diagnosis of hypertension. The first approach is to identify the socioeconomic, biological, and medical risk factors for hypertension. Many anthropometric, lifestyle, and laboratory indices have been proposed as risk factors for hypertension [7]. Multiple studies have evaluated interventions for subjects with several risk factors, such as body weight and alcohol consumption, and reported a significant lowering of BP and the incidence of hypertension [8,9].

The other approach is the development of a prediction model that measures the combined effects of risk factors [10,11,12]. Parikh et al. established a hypertension prediction model using approximately 1700 Americans enrolled in the Framingham Heart Study and the Framingham Offspring Study cohort of USA [10]. This study identified seven risk factors by Weibull regression and created a prediction model using these variables as input features [10]. A British team (Whitehall II study) constructed a hypertension prediction model based on about 6700 UK populations [11], and adopt the seven risk factors identified by Parikh et al. [10]. A study by Paynter et al. introduced a prediction model using the Women’s Health Study cohort of approximately 14800 Americans [12]. This model included the following nine predictors: age, ethnicity, systolic BP, diastolic BP, body mass index (BMI) and laboratory indices.

Hypertension has complicated pathophysiology interrelated to the cardiovascular system, renin-angiotensin system, autonomic nervous system, and other factors [13]. However, previous hypertension prediction models used only finger-countable risk factors. Therefore, these models cannot reflect the complex bio-signature of hypertension. This limitation, caused by the selection of only a few predictors, occurred because the statistical methods for selecting features had strict criteria when curating risk factors.

In this study, we performed three main tasks. First, we attempted to employ a large number of candidate variables using a literature-based selection of risk factors from demographic, medical, anthropometric, and laboratory variables. Second, we adopted weighting values estimated by KNHANES to increase the statistical power. Third, we constructed a sex-specific prediction model using a large-scale cross-sectional cohort and externally validated the model by longitudinal data.

## 2. Materials and Methods

### 2.1. Participants

Data from 2011 to 2018 KNHANES and the community-based Korean Genome and Epidemiology Study (KoGES) were utilized. KNHANES is a nationwide cross-sectional survey that includes health status, chronic disease prevalence, and nutrition status, and it is annually conducted by the Korea Centers for Disease Control and Prevention [14]. KoGES is an ongoing cohort study established by the Korean National Institute of Health, Korea Centers for Disease Control and Prevention in 2001 [15]. It contains a health and lifestyle survey, laboratory results, and chronic disease incidence in Korean adults. The current study used KoGES data up to December 2018.

We arranged the 2011–2015 KNHANES and 2016–18 KNHANES datasets as the derivation and internal validation (IV) sets for our hypertension prediction model, respectively. The KoGES dataset was used as an external validation (EV) set. We enrolled only those over 40 years of age. We excluded individuals in both KNHANES and KoGES with incomplete data on socioeconomic status, medical history, lifestyle-related variables, anthropometric data, and laboratory indices. Specifically, participants in KoGES were excluded when they were diagnosed with hypertension or had high BP (systolic BP ≥ 140 mmHg or diastolic BP ≥ 90 mmHg) at baseline. The final number of subjects was 15,395, 10,333, and 4633 in the derivation, IV, and EV sets, respectively.

All participants provided written informed consent prior to their participation in these KNHANES and KoGES surveys and their data were processed anonymously. The present study protocol was approved by the Institutional Review Board of Wonju Severance Christian Hospital (IRB No. CR320310).

### 2.2. Definition of Hypertension

Both in KNHANES and KoGES, hypertension was defined according to the Seventh Report of the Joint National Committee: (1) systolic BP ≥ 140 mmHg or diastolic BP ≥ 90 mmHg [16]; (2) previous diagnosis of hypertension by a physician; and (3) receiving antihypertensive medications.

### 2.3. Evaluation of Variables

We included education status, which was classified as elementary, middle, and high school and university graduation. Income status was categorized into quartiles.

Diabetes mellitus was defined as follows: (1) serum fasting glucose level ≥ 126 mg/dL [17]; (2) previously diagnosed by a physician; or (3) receiving glucose-lowering medications. Dyslipidemia was included for participants receiving lipid-lowering medications, and cancer was included based on a previous diagnosis by a physician.

We classified smoking habits into three stages: non-smoker, ex-smoker, and current smoker. We defined heavy alcoholics as the weekly consumption of over 70 g of alcohol in women and over 140 g in men. Nutritional information such as daily energy intake, was investigated based on a 24-h recall. Height, body weight, and waist circumference were measured in light clothes, and the BMI was calculated as weight/height^2^ (kg/m^2^). BP was recorded three times after resting for at least 5 min in a seated position [2,18]. The average value was selected after BP was measured using an appropriately sized arm cuff and a mercury sphygmomanometer.

Blood samples were collected after fasting for at least 8 h. These samples were immediately stored in a refrigerator and analyzed within 24 h. The levels of serum fasting glucose, creatinine, and lipid profiles were measured using a model 7600 auto-analyzer (Hitachi, Tokyo, Japan) in both surveys. White blood cell (WBC) count and hemoglobin (Hb) levels were measured with XE-2100D (Sysmex, Tokyo, Japan) in the KNHANES and ADVIA 120 hematology system (Bayer Corporation, Tarrytown, NY, USA) in KoGES [2,18].

### 2.4. Selection of Predictors

We determined predictive risk factors using two methods, literature-based search, and statistics-based selection, as reported previously [19]. First, two physicians with experience in treating hypertension searched the literature to select the candidate risk factors. Consequently, approximately 40 risk factors were reviewed, and detailed information about the individual studies is described in the Appendix A. Of the 40 risk factors, we included 19 variables, which were available in both KNHANES and KoGES.

We selected predictive candidates by backward-stepwise logistic regression (LR) after applying the weight values of the KNHANES. The weight value is a number indicating the number of people represented by the subject. When the KNHANES data was generated, this value was determined by the data constructor, considering the following factors: probabilities of selecting the participant, no-response adjustment, and post-stratification factors [20]. Therefore, the statistical results using the weight values adequately represent the Korean population. Note that the weight value was only used for the process of selecting the significant variables and constructing the prediction model. The candidate predictors, including smoking, alcohol consumption, total energy intake, fasting glucose, and triglyceride levels, were log-transformed in the feature-selection section. As the LR method has crucial limitations including multicollinearity, we used backward stepwise regression to minimize it.

### 2.5. Statistics

We used the Kolmogorov–Smirnov test to examine whether candidate variables, specifically continuous variables, were normally distributed. Continuous and categorical variables were analyzed based on hypertension status using Student’s *t*-test and Chi-square test, respectively. To test for linear trends of the candidate variables, we determined the median values of each probability quintile as continuous variables in the Chi-square test and one-way analysis of variance.

The hypertension classification model was constructed from the training dataset using an LR model. To validate the performance of the prediction model, we used the receiver-operating characteristic (ROC) curve and the area under the curve (AUC).

To address the overfitting or underfitting problems, we conducted following tasks. We used the weighted dataset (i.e., derivation dataset, KNHANES 2011–15) to establish the sex-specific hypertension classification model. We measured the prediction performance of the sex-specific model for the derivation dataset. In detail, we made 1000 datasets by bootstrapping the derivation dataset. We applied the model for the 1000 bootstrapped datasets, yielding the null distribution for the classification performance of the derivation dataset, including 1000 AUC values.

Cox regression analysis was used for the survival analysis. Statistical analysis was performed by R language (R package ver.4.0.1). Statistical significance was set at *p* value < 0.05.

To justify the predictors obtained from the LR model, we compared the LR model with the model based on Net Reclassification Improvement (NRI) method. For the NRI method, the determination of variables (i.e., standard variable) are needed to check whether the prediction performance is improved when a new variable is added. We set age, systolic BP, diastolic BP, smoking status, and BMI as the standard variables that were recommended by the previous study [10].

The variance importance for candidate hypertension-related features were measured using Classification And Regression Trees (CART). In detail, we implemented rpart function in rpart package to measure the degree of importance of candidate variables, and scaled the value of variance importance by following equation.
$$Scaled\;value=\frac{original\;value}{sum\;of\;other\;variable’s\;values}\times 100$$

## 3. Results

Table 1 and Appendix A summarize the general characteristics of the derivation and IV sets based on the status of hypertension and sex, respectively. The proportion of subjects with hypertension was 45.5% in men and 40.6% in women in the derivation dataset and 48.0% in men and 40.5% in women in the IV set. The mean age of the participants in the hypertension group was older than that of those in the non-hypertension group in both the derivation and IV datasets (Table 1 and Appendix A).

Participants with hypertension in both the derivation and IV sets had the following significantly different characteristics compared with those without hypertension: lower education levels, higher prevalence of chronic diseases including diabetes and dyslipidemia, lower total energy intake, higher levels of BMI and waist circumference, higher systolic BP and diastolic BP, higher serum levels of fasting glucose, creatinine, and WBC, and lower serum total cholesterol levels. The levels of income were lower and those of serum triglycerides and Hb were higher significantly in women with hypertension in both the derivation and IV datasets compared with those in women without hypertension. Furthermore, men with high BP smoked more, consumed more alcohol, and had significantly lower serum Hb levels than those without high BP in both datasets.

To screen the candidate features, we performed univariate LR by setting each predictor obtained from the literature search and binary form of hypertension status as independent and dependent variables, respectively. Consequently, we identified 19 risk factors with *p*-values < 0.05, which were subsequently used as candidate variables in multivariate LR in both men and women. Among those risk factors, 19 variables were all selected in Korean men (Model 2 of Table 2). In Korean women, 18 features were finally selected using backward stepwise LR except for smoking (Model 2 of Table 3). Therefore, we set 19 and 18 risk factors for hypertension as the input features in the classifying model in men and women, respectively.

We constructed a sex-specific hypertension classification model using the selected variables and LR. In our final hypertension classification model in men, the following variables were positively associated with hypertension: age, diabetes mellitus, dyslipidemia, alcohol consumption, BMI, waist circumference, systolic BP, diastolic BP, creatinine, triglycerides, and WBC levels. A negative relationship with hypertension was identified in income status, education status, cancer, smoking, total energy intake, serum fasting glucose, total cholesterol, and Hb level (Model 2 of Table 2). The results in women were similar to those in men, except for the following variables: smoking, waist circumference, serum fasting glucose, and Hb (Model 2 of Table 3).

Our model was based on the LR model that consisted of linear units (weighted sum equation) and non-linear units (referred to as activation function). In detail, non-linear units are sigmoid functions that convert the weighted sum results obtained from linear units to values ranging from zero to one as a similar form of probability. We measured the probability that subjects in the IV set had hypertension using the sex-specific model. Figure 1A,B present the receiver-operating characteristic (ROC) curve and their AUC in men and women, respectively. The classification performance for hypertension in Korean men was an AUC of 0.847 with the use of 19 risk factors. In the case of hypertension in Korean women, the performance was an AUC of 0.901 with the use of 18 risk factors.

We optimized the sex-specific hypertension classification model using the weighted samples. To check whether the model tends to be overfitting or underfitting, we measured the prediction performance of the sex-specific hypertension model for the derivation datasets. The prediction performance of IV dataset tended to be low in men (i.e., overfitting) than those of the resampled training datasets (Figure 2). In case of women, the hypertension classification model did not exhibit significant difference between performances of derivation and IV datasets.

In the KoGES dataset, subjects without hypertension or high BP at baseline were 2163 men and 2470 women. The mean follow-up duration of all the participants was 116.8 months (approximately 10 years). During the follow-up period, 847 and 855 cases of new-onset hypertension were noted in men and women, respectively. The average duration of new-onset hypertension was 79.0 months and 80.2 months in men and women, respectively. The baseline characteristics of the EV set in KoGES data are presented in the Appendix A, according to the presence or absence of new-onset hypertension. The following significant variables in participants with new-onset hypertension in the EV set were similar to those in participants with hypertension in the derivation set: age, education, BMI, waist circumference, systolic BP and diastolic BP in both men and women, alcohol consumption in men, and income status, serum triglyceride levels and Hb in women. Most of these features had the same direction between the EV set and the derivation set except for serum total cholesterol levels in women. Contrary to the characteristics of the derivation set, subjects with new-onset hypertension did not differ significantly from those without hypertension in the following variables in EV data: medical history of diabetes mellitus, dyslipidemia and cancer, smoking, total energy intake, and serum levels of fasting glucose, creatinine, and WBC in both men and women.

We calculated the probability of the participants in the KoGES (EV set) being diagnosed with hypertension currently, using the sex-specific model constructed from the derivation set (KNHANES). We divided the subjects into five groups based on the increasing order of probabilities. As the increment of the probabilities in Korean men, following variables had significant characteristics: older age, lower income, lower education, higher ratios of diabetes mellitus and dyslipidemia, more alcohol consumption, higher levels of BMI, waist circumference, systolic BP and diastolic BP, higher serum levels of fasting glucose, creatinine, triglyceride, and WBC, and lower serum level of total cholesterol (Appendix A). In case of Korean women, following differential characteristics were shown as the increment of probabilities: older age, lower income, lower education, higher ratio of diabetes mellitus, less alcohol consumption, less total energy intake, higher levels of BMI, waist circumference, systolic BP and diastolic BP, and higher serum levels of fasting glucose, creatinine, total cholesterol, triglyceride, WBC, and Hb (Appendix A).

Figure 3 and Figure 4 illustrate the cumulative incidence of new-onset hypertension and the *p*-value matrix. The *p* values in the matrix were Bonferroni-corrected *p* value [(nominal *p* value) × 10 (_5_C_2_)]. Based on the cut-off (Bonferroni-corrected *p* value < 0.05), most groups significantly differed from each other except for group 2 vs. 3 and 3 vs. 4 in men.

In addition, we validated the sex-specific hypertension model by measuring the classification performances for the new-onset hypertension in KoGES dataset (EV set). Figure 5 presents the AUCs of the sex-specific hypertension prediction model for the 2-, 6-, and 10-year new-onset hypertension. The AUCs of 2-, 6-, and 10-year follow-up were 0.718, 0.726, and 0.714 in men, and 0.777, 0.792, and 0.797 in women, respectively.

We implemented NRI to identify variables contributing the classification improvement with setting age, systolic BP, diastolic BP, smoking status, and BMI as standard variables that were recommended by the previous study [10]. We established the classification model by adding a variable (e.g., dyslipidemia, WBC count, and serum creatinine level) among the candidate predictors. We iterated the above works as many as the number of the candidate predictors. Then, we selected the variable with the highest NRI values among the candidate variables. As a result, we selected 14 and 8 features in men and women, respectively (Appendix A). We compared the hypertension prediction performances obtained from the model including predictors selected by the stepwise LR with those obtained from the model consisting of features curated by the NRI method. We found that the classification performance was slightly higher in the hypertension model including predictors selected by the stepwise LR than the model composing features curated by the NRI method (Figure 6).

Using CART analysis, we selected the top 10 contributing predictors in the sex-specific hypertension classification model. In the model for Korean men, the top three predictors were age, dyslipidemia, and systolic BP based on the variance importance. In case of Korean women, age, education, and systolic BP were the top three contributors. Age, dyslipidemia, systolic and diastolic BPs, education level, serum creatinine level, and waist circumference were top contributors for the hypertension classification in common between Korean men and women (Appendix A).

## 4. Discussion

We identified predictors using literature reviews and statistical methods for our hypertension classification model. The predictors included approximately 20 features that could reflect various pathophysiological aspects of hypertension. The hypertension classification model using the features extracted from the 2011–2015 KNHANES dataset accurately classified participants with hypertension in the 2016–2018 KNHANES dataset (IV set). Furthermore, we externally validated this model by predicting participants with new-onset hypertension in the longitudinal set (KoGES). A hypertension risk score model from the Framingham Heart Study had an AUC of 0.788 [10]. The KoGES model by Lim et al. resulted in AUCs of 0.791 and 0.790 via coefficient- and point-based scores, respectively [21]. A hypertension prediction model from the Atherosclerosis Risk in Communities Study and the Cardiovascular Health Study proposed AUCs of 0.751–0.754 for 3 years of follow-up and 0.773–0.776 for 9 years of follow-up [22]. Our model had AUCs of 0.847 and 0.901 in men and women for IV set, respectively, which are comparable with the performances of the other models.

For the input features of our model, we used 19 and 18 variables in men and women, respectively. The pathophysiology of hypertension is complicated [13]; therefore, diverse risk factors of hypertension have been reported (Appendix A). However, in previous studies, finger-countable clinical variables were included in the prediction models as a result of strict feature-selection methods [10,11,21,22]. Considering the results of our analysis, several variables in the longitudinal data (KoGES) demonstrated insignificant differences between participants with and those without hypertension. If we used the KoGES dataset to establish a prediction model with low statistical power (Appendix A), approximately 10 or fewer variables are expected to be selected, which is consistent with other prediction models [10,11,21,22]. To overcome this limitation, we used a large-scale cross-sectional study with weighted values instead of a relatively small longitudinal study. Consequently, we used approximately 20 various features that covered the socioeconomic status, medical history, lifestyle-related variables, anthropometric, and laboratory indices in the present study. Therefore, our prediction model included various pathophysiological aspects of hypertension. Furthermore, these variables can be easily collected in the clinical setting; therefore, physicians can apply this classification model in both real hospital-based and epidemiological studies.

Although several hypertension prediction models have been proposed, only a few studies have used sex-specific models [23,24,25]. Epidemiological studies have reported significant inter-sex differences in the prevalence and incidence of hypertension [26,27]. Considering the underlying mechanism of hypertension, women might have different hormonal effects compared to men in the development of hypertension, such as estrogen. Therefore, we constructed a sex-specific hypertension prediction model.

In our final hypertension classification model, convergent results with those of previous studies were observed in age and systolic BP in both men and women [10,11,21,22]. However, inconsistent associations with those of other studies were proposed in several variables, including smoking [10,11,21,22]. Smoking is a risk factor for cardiovascular disease [28], and a previous study demonstrated that it increases acute BP and pulse rate [29]. However, some researchers have reported that BP in current smokers was not significantly higher than that in non-smokers [30,31]. Furthermore, Primatesta et al. suggested that smoking is involved in BP in association with other factors such as alcohol consumption and BMI [30], rather than being an independent risk factor. In Table 1, alcohol consumption in women was significantly lower in hypertensive group than in non-hypertensive group, as well as total energy intake and serum levels of total cholesterol in both men and women. The identical results of total energy intake and serum levels of total cholesterol were presented in both men and women in Appendix A. The ambiguous causal relationship between hypertension and lifestyle indices derived from the analysis of the cross-sectional dataset might result in inconsistent or controversial findings, similar to our results.

There are several limitations to the present study. First, the dataset used for the prediction model was from a cross-sectional study. We utilized this dataset for two reasons: to include a large population and weighted values that were estimated by the data constructors. To overcome this limitation, we attempted to validate this model using a longitudinal dataset. Lee at al. [19] reported a study which created a suicide prediction model using a nationwide cross-sectional data, the Korea Youth Risk Behaviour Web-based Survey (KYRBS). In this study, authors selected 13 predictors by the statistical method of LR, and the resulting performance was an AUC of 0.85. Second, our results were only derived from the Korean population, thus yielding biased results for a specific ethnic group. Application of this model to different countries, races, or cultures may be difficult. Third, we implemented the LR algorithm as a classification model. LR consists of a weighted sum unit and a non-linear unit (sigmoid function) and has some crucial limitations, such as multicollinearity and inability to represent complex data. We used backward stepwise regression to minimize multicollinearity (Table 2 and Table 3). Furthermore, this method has been called a “shallow classifier” owing to its inability to represent complex data [32]. Consequently, deep learning can be used for prediction models that include a lot of input variables.

In conclusion, we proposed a hypertension risk model utilizing Korean datasets, which included data regarding risk factors that could be collected in clinical offices without difficulty. Physicians can utilize this model to predict the probability of hypertension in individuals and counsel high-risk individuals to modify their lifestyle and manage their diseases.

## Figures and Tables

**Figure 1 jpm-11-00968-f001:**
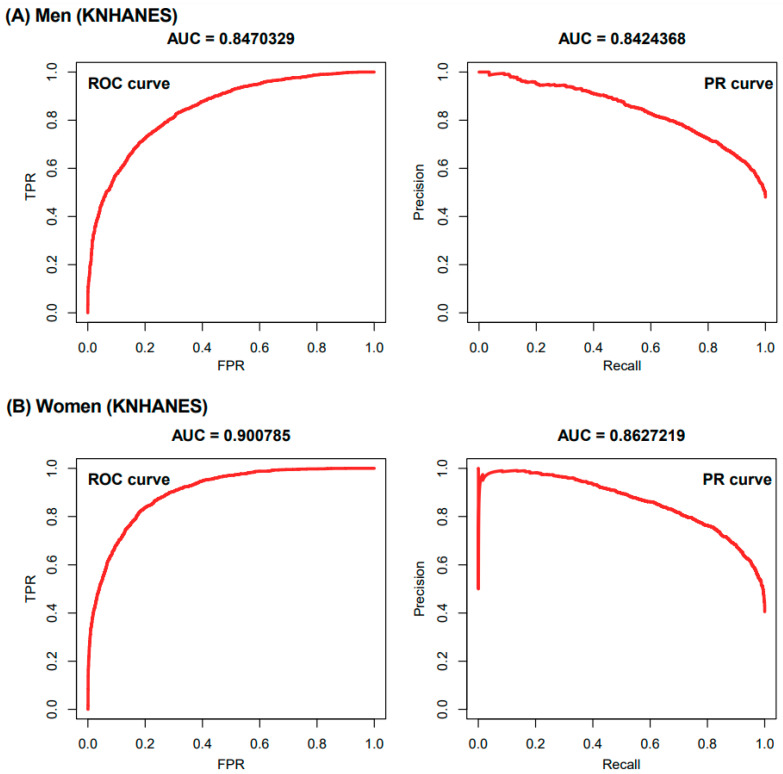
Receiver-operating characteristic curves for the sex-specific hypertension prediction model using KNHANES dataset from 2016 to 2018. (**A**) Men (AUC = 0.847) (**B**) Women (AUC = 0.901). Abbreviations: KNHANES, Korea National Health and Nutrition Examination Survey; ROC, receiver-operating characteristic; PR, precision-recall; AUC, area under the curve; TPR, true positive rate; FPR, false-positive rate.

**Figure 2 jpm-11-00968-f002:**
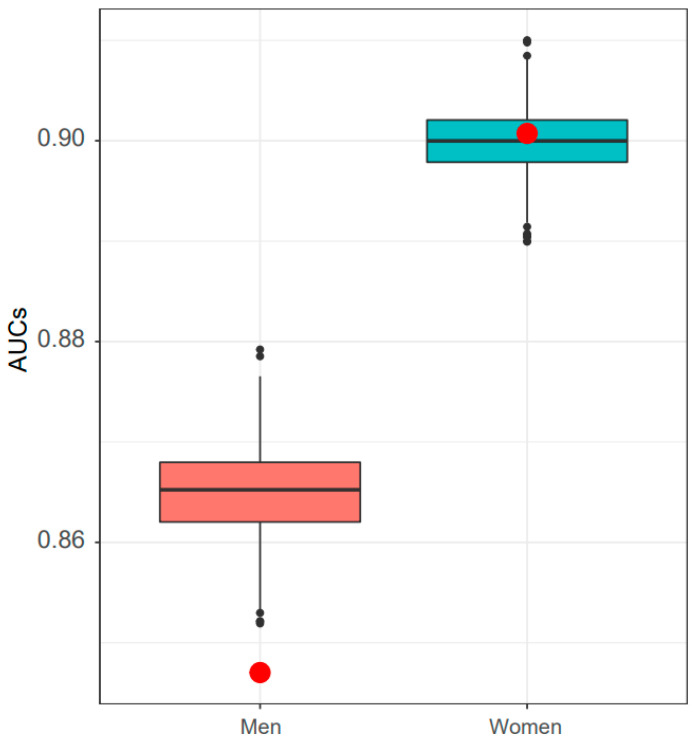
Classification performance of the sex-specific hypertension classification model. A boxplot includes 1000 bootstrapped derivation datasets (i.e., KNHANES 2011–2015). Red points indicate the classification performance obtained from the internal validation dataset (i.e., KNHANES 2016–2018). Abbreviations: AUC, area under the curve; KNHANES, Korea National Health and Nutrition Examination Survey.

**Figure 3 jpm-11-00968-f003:**
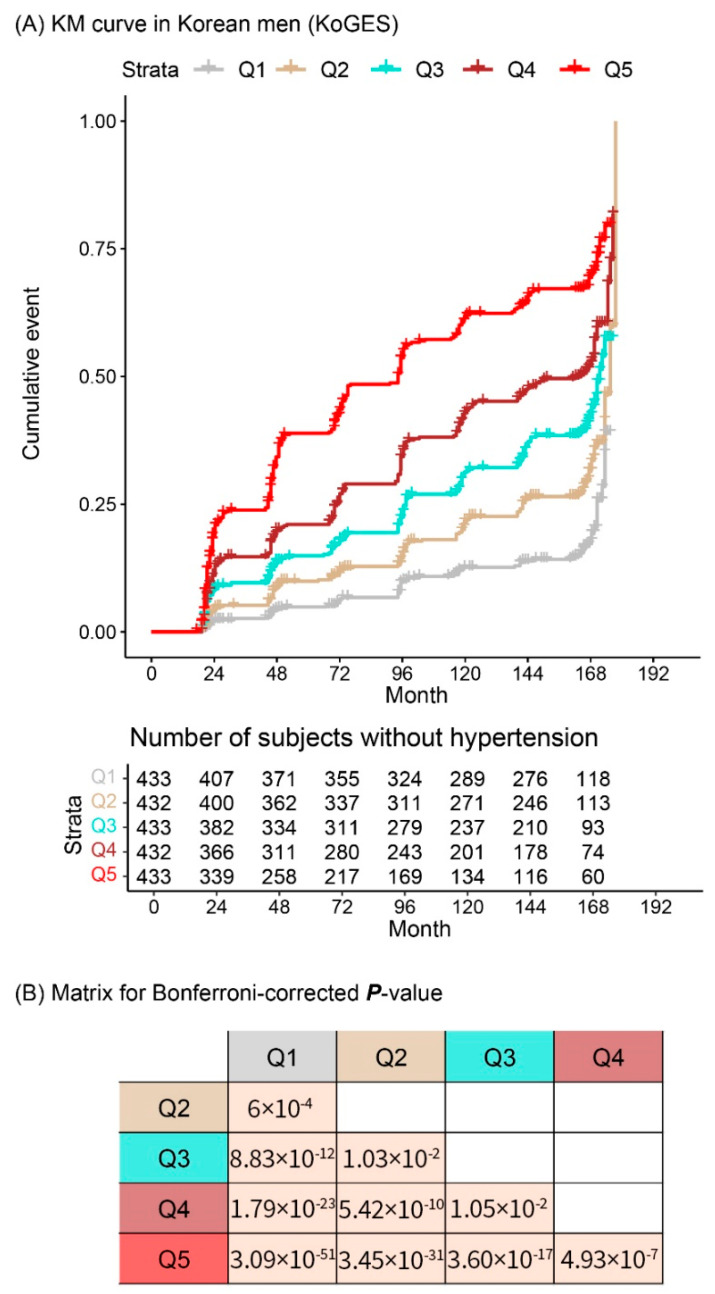
(**A**) Difference in cumulative incidences of new-onset hypertension between the five groups divided according to the expected probabilities in men in the KoGES study. (**B**) *p*-value matrix. Abbreviations: KM, Kaplan–Meier; KoGES, Korean Genome and Epidemiology Study; Q, Quintile.

**Figure 4 jpm-11-00968-f004:**
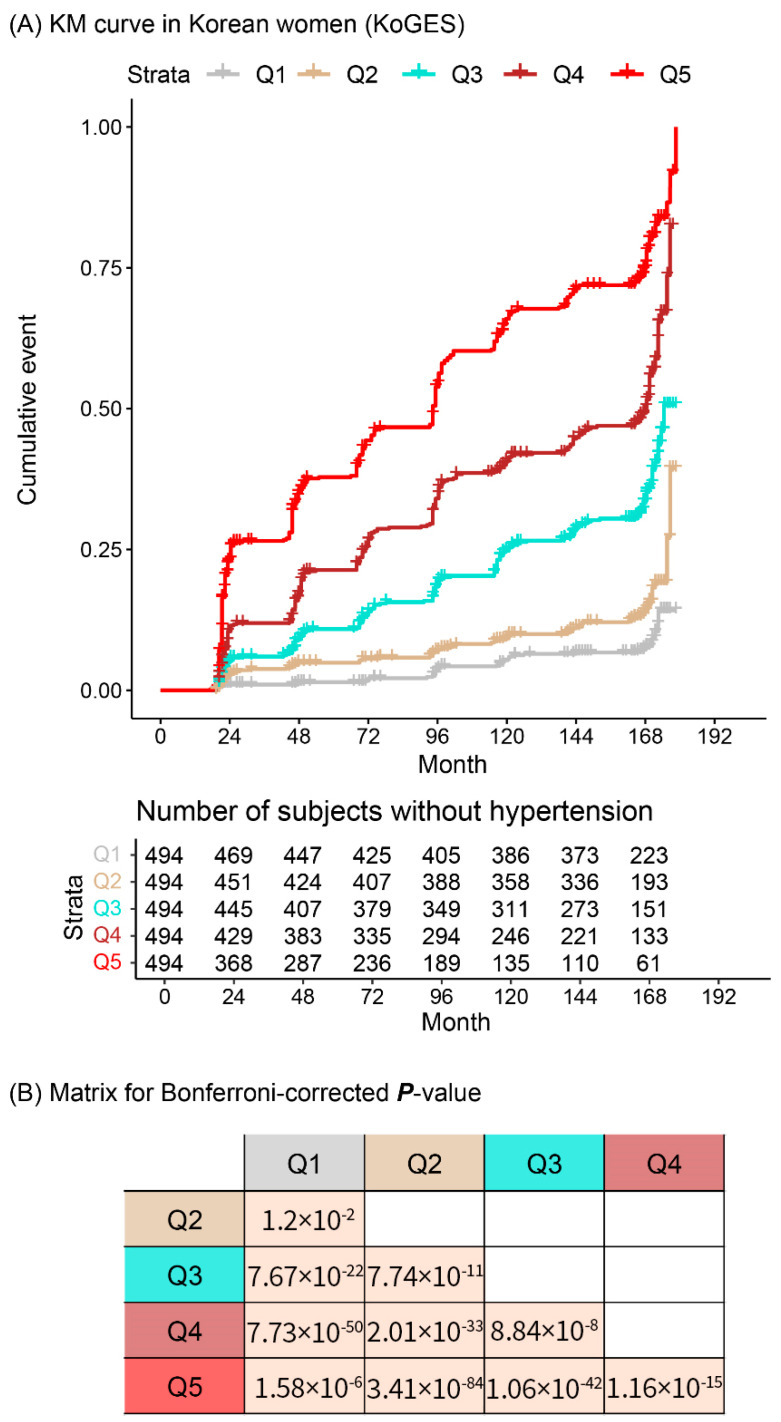
(**A**) Difference in cumulative incidences of new-onset hypertension between five groups divided according to the expected probabilities in women in the KoGES study. (**B**) *p*-value matrix. Abbreviations: KM, Kaplan–Meier; KoGES, Korean Genome and Epidemiology Study; Q, Quintile.

**Figure 5 jpm-11-00968-f005:**
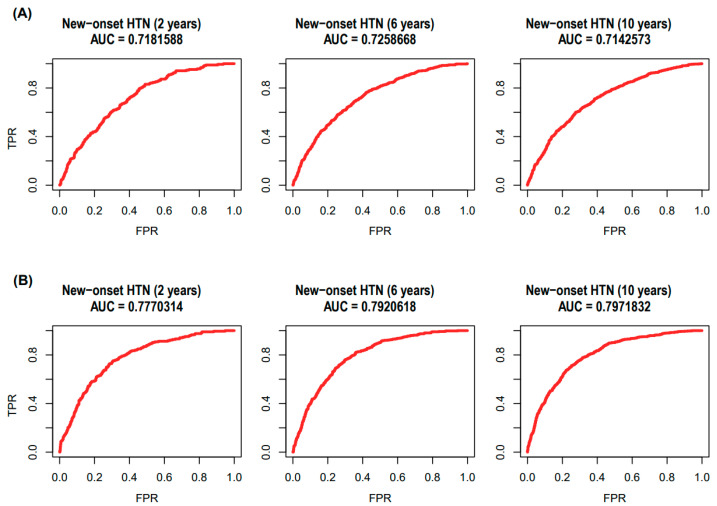
ROC curves for the sex-specific hypertension prediction model using KoGES dataset. (**A**) New-onset HTN during 2-year follow-up (AUC = 0.718), 6-year follow-up (AUC = 0.726), and 10-year follow-up (AUC = 0.714) in Men. (**B**) New-onset HTN during 2-year follow-up (AUC = 0.777), 6-year follow-up (AUC = 0.792), and 10-year follow-up (AUC = 0.797) in women. Abbreviations: ROC, receiver-operating characteristic; KoGES, Korean Genome and Epidemiology Study; AUC, area under the curve; HTN, hypertension; TPR, true positive rate; FPR, false-positive rate.

**Figure 6 jpm-11-00968-f006:**
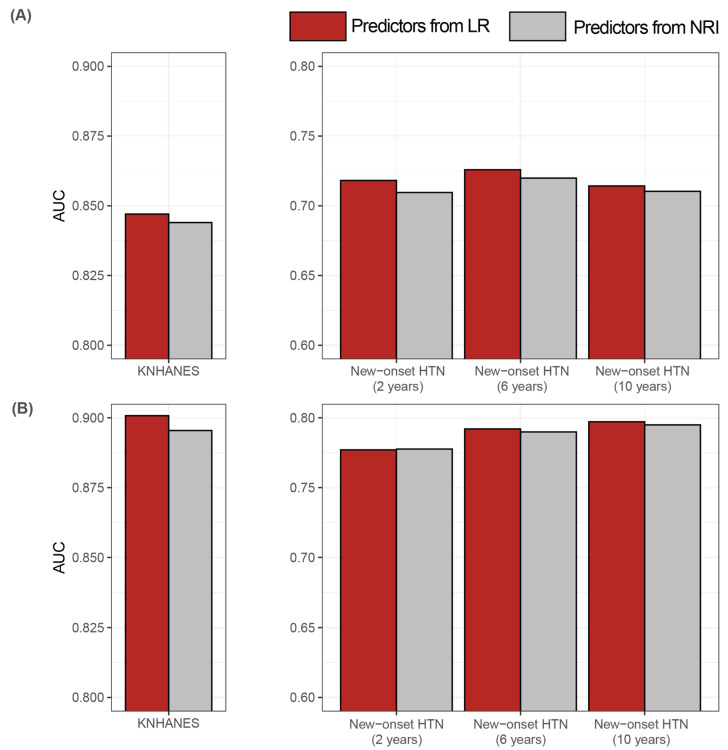
Classification performances of the sex-specific hypertension classification models consisting of predictors selected by stepwise LR and NRI using KNHANES and KoGES datasets. (**A**) Men (**B**) Women. Abbreviations: LR, logistic regression; NRI, net reclassification improvement; KNHANES, Korea National Health and Nutrition Examination Survey; KoGES, Korean Genome and Epidemiology Study; AUC, area under the curve; HTN, hypertension.

**Table 1 jpm-11-00968-t001:** General characteristics of derivation dataset (KNHANES 2011–15).

	Derivation Dataset (KNHANES 2011–15) *n* = 15,395					
	Men			Women		
	Non-Hypertension	Hypertension	*p* Value	Non-Hypertension	Hypertension	*p* Value
	*n* = 3472	*n* = 2900		*n* = 5364	*n* = 3659	
Age, years	57.5 ± 0.18	63 ± 0.19	<0.001	55 ± 0.14	64.8 ± 0.16	<0.001
Income, *n*			0.788			<0.001
1st quartile	798 (23)	690 (23.8)		1244 (23.2)	928 (25.4)	
2nd quartile	893 (25.7)	767 (26.4)		1279 (23.8)	999 (27.3)	
3rd quartile	841 (24.2)	747 (25.8)		1380 (25.7)	891 (24.4)	
4th quartile	940 (27.1)	696 (24)		1461 (27.2)	841 (23)	
Education, *n*			<0.001			<0.001
Elementary school	731 (21.1)	850 (29.3)		1519 (28.3)	2206 (60.3)	
Middle school	510 (14.7)	486 (16.8)		757 (14.1)	534 (14.6)	
High school	1116 (32.1)	938 (32.3)		1934 (36.1)	681 (18.6)	
University	1115 (32.1)	626 (21.6)		1154 (21.5)	238 (6.5)	
Diabetes mellitus, *n*	462 (13.3)	743 (25.6)	<0.001	365 (6.8)	823 (22.5)	<0.001
Dyslipidemia, *n*	149 (4.3)	443 (15.3)	<0.001	370 (6.9)	838 (22.9)	<0.001
Cancer, *n*	125 (3.6)	92 (3.2)	0.385	249 (4.6)	185 (5.1)	0.394
Smoking, pack-years	20.4 ± 0.34	21.9 ± 0.39	0.004	0.6 ± 0.05	0.8 ± 0.08	0.073
Alcohol consumption, g/week	99.7 ± 2.66	122.5 ± 3.16	<0.001	17.2 ± 0.77	13.5 ± 0.9	0.001
Total energy intake, kcal	2348.1 ± 16.33	2189.6 ± 15.68	<0.001	1725.7 ± 9.26	1582.8 ± 10.16	<0.001
BMI, kg/m^2^	23.6 ± 0.05	24.6 ± 0.06	<0.001	23.4 ± 0.04	25 ± 0.06	<0.001
Waist circumference, cm	84 ± 0.14	87.4 ± 0.16	<0.001	78.7 ± 0.12	84 ± 0.15	<0.001
Systolic BP, mmHg	115.5 ± 0.19	133.1 ± 0.31	<0.001	113 ± 0.17	134.8 ± 0.28	<0.001
Diastolic BP, mmHg	75 ± 0.14	80.6 ± 0.23	0.001	72.5 ± 0.1	78.5 ± 0.19	<0.001
FPG, mg/dL	102.8 ± 0.42	109.3 ± 0.5	<0.001	96.8 ± 0.28	105.5 ± 0.42	<0.001
Creatinine, mg/dL	0.99 ± 0.0027	1.01 ± 0.0098	<0.001	0.7 ± 0.0021	0.8 ± 0.0052	<0.001
Total cholesterol, mg/dL	190.9 ± 0.59	182.9 ± 0.67	<0.001	198 ± 0.48	195.5 ± 0.63	0.001
Triglyceride, mg/dL	152.8 ± 2.12	165.2 ± 2.33	0.762	117.7 ± 0.98	144 ± 1.49	<0.001
WBC, thousand/μL	6.4 ± 0.03	6.6 ± 0.03	<0.001	5.6 ± 0.02	6.1 ± 0.03	<0.001
Hb, g/dL	15.1 ± 0.02	14.9 ± 0.03	<0.001	13.1 ± 0.02	13.2 ± 0.02	<0.001

Continuous variables are presented as mean ± standard error, and categorical variables are presented as numbers (percentage, %). Abbreviations: KNHANES, Korea National Health and Nutrition Examination Survey; BMI, body mass index; BP, blood pressure; FPG, fasting plasma glucose; WBC, white blood cells; Hb, hemoglobin.

**Table 2 jpm-11-00968-t002:** Risk factors for hypertension selected by backward-stepwise logistic regression in Korean men.

	Model 1 (Univariate LR)	Model 2 (Multivariate LR)
	OR (95% CI)	OR (95% CI)
Age (years)	1.047 (1.047–1.047)	1.065 (1.065–1.066)
Income status (Ref: Q1)	0.975 (0.972–0.978)	0.94 (0.936–0.944)
Education status (Ref: Elementary)	0.782 (0.78–0.784)	0.961 (0.956–0.965)
Diabetes mellitus	2.237 (2.218–2.257)	1.494 (1.469–1.519)
Dyslipidemia	4.086 (4.032–4.14)	4.398 (4.324–4.474)
Cancer	0.937 (0.918–0.957)	0.633 (0.616–0.651)
Smoking (pack-years)	1.038 (1.037–1.04)	0.957 (0.955–0.96)
Alcohol consumption (g/week)	1.072 (1.071–1.073)	1.077 (1.075–1.079)
Total energy intake (kcal)	0.792 (0.788–0.797)	0.939 (0.931–0.947)
BMI (kg/m^2^)	1.139 (1.138–1.141)	1.087 (1.084–1.09)
Waist circumference (cm)	1.052 (1.051–1.052)	1.013 (1.012–1.014)
Systolic BP (mmHg)	1.117 (1.116–1.117)	1.091 (1.091–1.092)
Diastolic BP (mmHg)	1.083 (1.083–1.084)	1.065 (1.065–1.066)
FPG (mg/dL)	2.886 (2.851–2.92)	0.86 (0.84–0.879)
Creatinine (mg/dL)	4.368 (4.281–4.456)	6.453 (6.28–6.63)
Total cholesterol (mg/dL)	0.996 (0.995–0.996)	0.993 (0.993–0.993)
Triglyceride (mg/dL)	1.251 (1.246–1.256)	1.094 (1.087–1.1)
WBC (thousand/μL)	1.076 (1.074–1.078)	1.076 (1.073–1.079)
Hb (g/dL)	0.964 (0.961–0.966)	0.86 (0.856–0.863)

Abbreviations: BMI, body mass index; BP, blood pressure; FPG, fasting plasma glucose; WBC, white blood cells; Hb, hemoglobin; LR, logistic regression; OR, odds ratio; CI, confidence interval.

**Table 3 jpm-11-00968-t003:** Risk factors for hypertension selected by backward-stepwise logistic regression in Korean women.

	Model 1 (Univariate LR)	Model 2 (Multivariate LR)
	OR (95% CI)	OR (95% CI)
Age (years)	1.094 (1.093–1.094)	1.064 (1.063–1.065)
Income status (Ref: Q1)	0.933 (0.93–0.936)	0.991 (0.987–0.995)
Education status (Ref: Elementary)	0.495 (0.494–0.497)	0.868 (0.864–0.873)
Diabetes mellitus	4.024 (3.984–4.064)	1.798 (1.766–1.831)
Dyslipidemia	4.191 (4.149–4.234)	2.68 (2.643–2.717)
Cancer	1.143 (1.126–1.161)	0.945 (0.926–0.965)
Smoking (pack-years)	0.997 (0.993–1.001)	-
Alcohol consumption (g/week)	0.914 (0.913–0.915)	1.019 (1.017–1.021)
Total energy intake (kcal)	0.691 (0.688–0.695)	0.938 (0.93–0.945)
BMI (kg/m^2^)	1.165 (1.164–1.166)	1.104 (1.101–1.107)
Waist circumference (cm)	1.068 (1.068–1.068)	0.991 (0.99–0.992)
Systolic BP (mmHg)	1.117 (1.117–1.118)	1.09 (1.09–1.091)
Diastolic BP (mmHg)	1.083 (1.082–1.083)	1.044 (1.043–1.045)
FPG (mg/dL)	6.414 (6.324–6.505)	1.192 (1.165–1.22)
Creatinine (mg/dL)	6.207 (6.053–6.365)	4.679 (4.522–4.842)
Total cholesterol (mg/dL)	0.999 (0.999–0.999)	0.995 (0.995–0.995)
Triglyceride (mg/dL)	1.734 (1.726–1.741)	1.109 (1.102–1.116)
WBC (thousand/μL)	1.18 (1.178–1.182)	1.064 (1.061–1.067)
Hb (g/dL)	1.149 (1.146–1.153)	1.013 (1.009–1.017)

Abbreviations: BMI, body mass index; BP, blood pressure; FPG, fasting plasma glucose; WBC, white blood cells; Hb, hemoglobin; LR, logistic regression; OR, odds ratio; CI, confidence interval.

## Data Availability

The data of KNHANES is publicly available at https://knhanes.kdca.go.kr/knhanes/eng/index.do, accessed on 7 April 2020, and the data of KoGES can be provided after evaluation of research plan by the Korea National Institute of Health, Korea Centers for Disease Control and Prevention (http://www.nih.go.kr/contents.es?mid=a50401010100, accessed on 21 April 2020).

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
