# Peer review of "Development of the Hypertension Index Model in General Adult Using the Korea National Health and Nutritional Examination Survey and the Korean Genome and Epidemiology Study"

_jpm, 2021, doi:10.3390/jpm11100968_

Round 1
Reviewer 1 Report
Thank you for the opportunity to let me review the manuscript "Development of the Hypertension Index Model in General Adult Using the
Korea National Health and Nutritional Examination Survey and the Korean
Genome and Epidemiology Study". For developing the hypertension prediction model, the one of the strengths of this study is utilized the population-based data with using weighted value to represent the population. However, my major concern on the manuscript is the data and the study design.
1. As the authors already described in the manuscript, the KNHANES data is cross-sectional data. One of the major limitations of the cross-sectional data is hard to determine the casual relationship. Nevertheless, the authors used the data for deriving the risk factors for hypertension, and for the inter-validation. I think this is why the AUCs could be higher than other references.
Although the sample number is small, I would like the authors to analyze the ROC and AUC from the KoGES data using developed models.
(If the AUC is low from the analysis, then the authors could find other data to validate the model.)
2. Moreover, the authors defined the hypertension as the participants who diagnosed with hypertension (by doctor) in KNHANES data.
If the authors considered the causality, the authors should have included participants who had never been diagnosed with hypertension in their lifetime. Instead of previous definition that the authors used, hypertension can be defined by systolic BP and diastolic BP. The variables are available in the KNHANES data.
3. If there is any previous study that used cross-sectional data to develop prediction model, please cite and explain regarding using the cross-sectional data after the line number 326, page 12.
4. The Figure 1 has a typo. "(A) Men (KNHANES)" was written twice in different images.
5. In KoGES data, the authors categorized participants into five groups based on the increasing order of probabilities using the prediction models. Please provide the probability range of each group and distribution of the general characteristics of this five groups as supplementary data, instead of Table S3.
Reviewer 2 Report
The authors have developed a model to predict hypertension using clinical variables that could be easily collected in a clinical setting. Although the current version reads well, I am not convinced with the method the predictors were added. I believe it is an overfitted model in addition to some predictors being added based on literature. Would recommend the authors to justify the model based on NRI. Below are my comments.
- The model seem to contain 19 predictor variables. How did the authors address the overfitting of the model? How was collinearity handled? Could the authors please elaborate in the statistical section?
- Would recommend the authors to report the net reclassification index for each predictor. It would be more useful to select the best model based on NRI. This would help the authors to justify the predictors added to the model.
- Could the authors provide a CART analysis flowchart that demonstrates the contribution of each predictor variable?
Round 2
Reviewer 1 Report
I am appreciate with the authors' sincere response. I would like to give one last comment for this study.
-Some variables, such as education status, is hard to change before or after the onset of hypertension. However, habitual variable such as smoking or alcohol consumption could be change to improve one's health after diagnosis of hypertension by a doctor.
Because this study is for the prediction model, and the study used cross-sectional data for AUC calculation, bias should be minimized by handling the data. Therefore, I think participants who were diagnosed with hypertension by the doctor, or who were prescribed the hypertension medications must be excluded in the data to minimize the bias.
I would like the authors to additionally analyze the ROC curve and calculate the AUC after excluding the participants who were diagnosed with hypertension by the doctor or who were prescribed the hypertension medications in KNHANES data. If results are not differ from the current results described in the revised version of the manuscript, I think it is fair to take the current results.
Reviewer 2 Report
I thank the authors for appropriately addressing all the comments. Happy for the manuscript to be accepted in its current form.